# Transcriptomic Analysis of Planarians under Simulated Microgravity or 8 g Demonstrates That Alteration of Gravity Induces Genomic and Cellular Alterations That Could Facilitate Tumoral Transformation

**DOI:** 10.3390/ijms20030720

**Published:** 2019-02-08

**Authors:** Nídia de Sousa, Gustavo Rodriguez-Esteban, Ivan Colagè, Paolo D’Ambrosio, Jack J. W. A. van Loon, Emili Saló, Teresa Adell, Gennaro Auletta

**Affiliations:** 1Department of Genetics, Microbiology and Statistics, and Institute of Biomedicine, University of Barcelona, Catalonia, 08028 Barcelona, Spain; nidia.desousa@gmail.com (N.d.S.); esalo@ub.edu (E.S.); 2CNAG-CRG, Centre for Genomic Regulation (CRG), Barcelona Institute of Science and Technology (BIST), 08028 Barcelona, Spain; gustavo.rodriguez@cnag.crg.eu; 3Universitat Pompeu Fabra (UPF), 08002 Barcelona, Spain; 4Pontifical University Antonianum, via Merulana 124, 00185 Rome, Italy; i.colage@antonianum.eu (I.C.); paolodam.prof@gmail.com (P.D.); 5Pontifical University of the Holy Cross, DISF Centre, Via dei Pianellari 41, 00186 Rome, Italy; 6University of Cassino, Via Zamosch 43, 03043 Cassino, Italy; 7Dutch Experiment Support Center (DESC), Department of Oral and Maxillofacial Surgery/Oral Pathology, Amsterdam UMC location VU University Medical Center & Academic Centre for Dentistry Amsterdam (ACTA), 1081 LA Amsterdam, The Netherlands; j.vanloon@vumc.nl; 8European Space Agency-ESA-Technology Center-ESTEC, TEC-MMG-Lab, 2200 AG Noordwijk, The Netherlands; 9Pontifical Gregorian University, Piazza della Pilotta 4, 00187 Roma, Italy

**Keywords:** development, hypergravity, human health, microgravity, planarian, regeneration, space exploration, transcriptomics, Random Positioning Machine (RPM), Large Diameter Centrifuge (LDC)

## Abstract

The possibility of humans to live outside of Earth on another planet has attracted the attention of numerous scientists around the world. One of the greatest difficulties is that humans cannot live in an extra-Earth environment without proper equipment. In addition, the consequences of chronic gravity alterations in human body are not known. Here, we used planarians as a model system to test how gravity fluctuations could affect complex organisms. Planarians are an ideal system, since they can regenerate any missing part and they are continuously renewing their tissues. We performed a transcriptomic analysis of animals submitted to simulated microgravity (Random Positioning Machine, RPM) (s-µg) and hypergravity (8 g), and we observed that the transcriptional levels of several genes are affected. Surprisingly, we found the major differences in the s-µg group. The results obtained in the transcriptomic analysis were validated, demonstrating that our transcriptomic data is reliable. We also found that, in a sensitive environment, as under Hippo signaling silencing, gravity fluctuations potentiate the increase in cell proliferation. Our data revealed that changes in gravity severely affect genetic transcription and that these alterations potentiate molecular disorders that could promote the development of multiple diseases such as cancer.

## 1. Introduction

Planarians, freshwater flatworms, are bilaterally symmetric metazoans of the phylum *Platyhelminthes*. They possess well-defined antero-posterior and dorso-ventral axes, with an anterior cephalic region containing the brain and a pair of eyespots, a central region with the pharynx and the mouth opening, and a posterior tail region [1,2]. Planarians have the ability to perfectly regenerate complete new animals, from almost any fragment of their body, just in a few days. This amazing ability, that has caught scientist’s interest since long time ago, is due to the presence of a unique population of adult totipotent stem cells, known as neoblasts [3,4]. Planarians possess complex organs as central nervous system, which conserves all neural cell types and neuropeptides found in vertebrates [5]. These features make planarians the perfect model organism to perform molecular analysis to deeply understand animal regeneration and stem cell biology.

The possibility of space travel is growing. In several decades, inter-planetary trips will become common. However, the effects of gravity fluctuations on human health are only partially understood. Environmental factors continuously affect gene expression determining the phenotypical configuration and behavior of organisms, while gravity is the most constant environmental parameter influencing life evolution on Earth [6]. Modifications of gravity conditions can significantly affect phenotypic formation and differentiation [7,8]. At the organism level, it has been observed that exposure to microgravity can have consequences on key environment-sensitive processes like reproduction, development, and aging [9]. At the cellular level, it has been shown that microgravity alters the structure of cells, as well as their apoptotic and mitotic responses [6,7,8]. Hypergravity also has a relevant impact in cells behavior and gene expression [10,11,12,13]. However, specific modifications in gene expression, gene products, and related structural-functional phenotypic diversification due to sustained changes in gravity conditions remain poorly understood. To shed light into the consequences of more chronic gravity fluctuations in a complete organism, we used planarians as a model system. The main goal of our research was to analyze changes in gene expression on the basis of transcriptomic analysis in regenerating planarians, in simulated altered gravity by comparing samples exposed to simulated microgravity (s-µg) and hypergravity (8 g). Our results demonstrate that s-µg and hypergravity significantly modify gene expression in planarians, and that this effect could have an impact on cell behavior and also in synergy with sensitized backgrounds, since *hippo (RNAi)* animals maintained in s-µg or 8 g show a higher increase in cell proliferation. These results showed that prolonged exposure to an extra-terrestrial environment, like the s-µg environment during human space missions, could have severe consequences at genetic level, which could impact in promoting diseases such as cancer. 

## 2. Results

### 2.1. Simulated Microgravity and Hypergravity Change Planarian Gene Expression

Planarians were loaded in the Random Positioning Machine (RPM) to simulate µg, or in the Large Diameter Centrifuge (LDC) to apply 8 g gravity (Figure 1). The day after, planarians were cut at pre- and post-pharyngeal level. The transcriptome of those animals was analyzed 5 and 12 days after the amputation (i.e., after 6 and 13 days of s-µg or 8 g exposure, respectively), and compared to their corresponding controls maintained at normal gravity (1 g). The results of the differential expression analysis are shown in Appendix A. Sample-to-sample distances were calculated using Pearson correlation (Figure 2A), showing a better correspondence between those of the same regeneration stage—at 5 or 12 days of regeneration (dR). In agreement with that, principal component analysis (PCA) grouped the samples by time of regeneration in two main clusters (Figure 2B). In addition, within each group, most of the samples corresponding to the same gravity conditions were also clustered together. 

The particular transcripts found up- or down- regulated in each condition are shown in Appendix A. Our results demonstrate that after 5 dR (six days of s-µg or 8 g exposure), there are minor differences between samples (Figure 2C). Conversely, at 12 dR (13 days of s-µg or 8 g exposure), we found several genes deferentially expressed in exposed animals with respect to their corresponding controls (Figure 2C). Interestingly, the number of differentially expressed genes was much higher in animals regenerating in s-µg conditions than in animals regenerating at 8 g (Figure 2C).

Genes related with extracellular matrix (morphogenesis, cell differentiation), actin cytoskeleton (cell migration, proliferation), transposable elements and genes involved in stress response, like chaperones and heat shock proteins, were found down-regulated in s-µg conditions (Appendix A). At 8 g, there are fewer genes up- or down-regulated compared with s-µg conditions, and, interestingly, many of them cannot be attributed to any known class (40).2% at 8 g versus 25.9% at s-µg) (Appendix A). Down-regulated genes at 8 g include genes related to microtubules, cell communication and cell cycle. Finally, Dynein, which is a protein required for axonal transport and is involved in neurodegenerative diseases like Alzheimer [14], is found to be significantly down-regulated both in s-µg and at 8 g. 

The analysis of gene ontology (GO) functions associated to transcripts significantly up- and down-regulated at each condition (Appendix A), showed that genes related with biogenesis, namely ribosomal activity and transcription/translation of proteins and metabolic processes—such as mitochondrial proteins—were affected both under s-µg and 8 g conditions (Figure 2D). Since we found the major differences in gene transcription at 12 dR under s-µg conditions, we performed GO enrichment analysis on this data (Figure 2E, Appendix A). When focusing on the biological process, the main classes of up-regulated genes in these conditions include genes related with metabolic and biosynthetic pathways. Specific genes found in the transcriptome falling in these categories are ribosomal genes and genes with a role in translation (*sui1* factor) (Appendix A). This result is interesting, since dysregulated messenger RNA translation and specifically ribosome biogenesis is linked to the development and progression of hematological and metabolic diseases and cancer [15]. Other genes de-regulated at 12 dR and s-µg conditions are also involved in tumoral transformation. H^+^-ATP synthase is a conserved hub in mitochondria-to-nucleus signaling controlling cell fate and its deregulation contributes to cancer origin and progression [16]. The DNA repair protein rad50 is already down-regulated at 5 dR in s-µg conditions (Appendix A). DNA repair is an essential step in avoiding genetic instability and cancer development [17].

Importantly, despite the evident deregulation of gene expression, planarians trunks exposed to s-µg or 8 g regenerated an apparently perfect head (Figure 3).

### 2.2. Validation of Candidate Genes Found in the Differential Expression Analysis

To validate the results of the transcriptomic analysis, we performed a second experimental design. To increase the differences between conditions (s-µg and 8 g), head fragments were used instead of trunk fragments, since they are smaller and must regenerate the rest of the body, making the regeneration process more challenging. Head fragments were loaded in the RPM to simulate microgravity, in the same conditions as in the previous experiment (Figure 4A). After six days, regenerated animals were fixed and processed for in situ hybridization, in order to determine the levels of expression of selected candidate genes from the transcriptome. Five candidate genes which were down-regulated with respect to 1 g, were selected from the 12 dR group submitted to s-µg (Figure 2C and Figure 4B). The first gene was *collagen-α-1,* since it is an essential protein of the extracellular matrix [18], and the GO enrichment analysis indicated that collagen trimer was down-regulated at s-µg (Appendix A). We also selected *ADAMTS-like-1*, which codifies for a secreted protein involved in cell-cell signaling [19], *antigen Ki-67*, required for cell proliferation [20], and *piwi-1* and -*2*, two RNA binding proteins essential for the maintenance and function of stem cells [21]. The results show that three of them, *collagen* and *piwi-1* and -*2* are down-regulated in these animals after six days of exposure (Figure 4C), in agreement with the results found in the transcriptomic analysis of the previous experiment (Appendix A). We found that *ADAMTS-like-1* was up-regulated after 6 days of s-µg, in contrast with transcriptomic results, where it is down-regulated (Figure 4C). Several reasons could account for this result. For instance, the fragment analyzed in this experiment is a regenerating head instead of a trunk and different signaling pathways can be activated in the regeneration of different fragments since the structures to restore are different. In addition, the second experiment was performed in 6 dR animals, instead of 12 dR animals, the time-point at which this gene was found down-regulated in the transcriptomic analysis. The antigen *Ki-67* was not detected by in situ hybridization, probably because its low expression levels, since in situ hybridization is not as sensitive as RNA-seq analyses. 

These results allow us to corroborate the importance of transcriptomic approach to analyze the genetic changes in different conditions and confirm that planarians in s-µg conditions show deregulation of essential genes, such as *collagen*, despite their apparently normal regeneration. Moreover, the finding that *piwi* genes are down-regulated indicates that the population of neoblasts, which are the stem cells underlying planarian regenerative properties, are affected.

### 2.3. Gravity Changes in Hippo Silenced Planarians Promote Cell Proliferation

The transcriptomic data revealed that the expression of metabolic and biosynthesis genes is up-regulated in both s-µg and 8 g conditions (Figure 2, Appendix A). Since metabolic reprogramming has been recognized as one of the 10 cancer hallmarks essential for tumoral progression [22], these data suggest that gravity fluctuations could facilitate the tumor transformation. To test this hypothesis, we submitted uninjured planarians in which the Hippo pathway had been previously silenced by RNAi to s-µg and 8 g. The Hippo pathway is a well-known signaling pathway that controls the transcription of genes related with cell cycle, cell death and differentiation in response to the physical properties of the cellular environment (cell-cell or cell-matrix contact) [23,24]. In several human cancers, the Hippo pathway is found to be down-regulated [25] and its inhibition promotes overgrowth formation in multiple model organisms [26], including planarians [24].

In this experiment we analyzed the proliferative rates in *gfp (RNAi)* (control) and *hippo (RNAi)* (*hippo* dsRNA injected) planarians which were maintained in s-µg or 8 g conditions for 10 days (Figure 5A). Their respective experimental controls maintained at 1 g were also analyzed. At the end of the experiment, planarians were fixed and processed for immunohistochemical analysis using anti-PH3, to quantify mitotic activity. The results show an increase in mitotic cells in *hippo (RNAi)* animals maintained at 1 g (Figure 5B), as previously reported [24]. This increase in mitotic cells in *hippo (RNAi)* animals was observed in both control conditions, in the animals maintained at 1 g in the RPM and in the animals maintained at 1 g in the LDC. Importantly, the quantification of the mitotic activity of *hippo (RNAi)* animals maintained in s-µg, demonstrates that proliferation was increased with respect to the *hippo (RNAi)* animals maintained at 1 g and with respect to the *gfp (RNAi)* animals maintained at s-µg (Figure 5B). Planarians at 8 g show an increase of the mitotic activity compared to their control at 1 g. Importantly, the group exposed to 8 g in which *hippo* was silenced showed even a higher increase in the number of mitotic cells with respect with respect to the *hippo (RNAi)* animals maintained at 1 g and with respect to the *gfp (RNAi)* animals maintained at 8 g (Figure 5B). Therefore, our data demonstrates that gravity fluctuations in a situation of uncontrolled proliferation and growth, like the one produced by *hippo (RNAi)*, significantly intensify these processes. 

## 3. Discussion

The present study demonstrates that changes in gravity produce several genetic alterations in planarians. Although these alterations could have no apparent consequences, e.g., during the regeneration of new tissues in planarians, our data shows that during long exposures or in a sensitized background, they could lead to transcriptional changes, cellular malfunctions (over-proliferation) and progress of some diseases such cancer.

Concerning the reported genetic alterations, it is interesting to note that, although after six days of exposure to altered-gravity conditions only a few genes appear deregulated, and after 13 days, the number of deregulated genes drastically increases, supporting the long-term effects of altered gravity. In fact, this could be a reason why planarians regenerate correctly, since during the first days of the process, that are the crucial ones for regenerating new structures, the genetic alterations are minimal. Furthermore, a much higher number of deregulated genes (720 versus 77) appear in s-µg conditions with respect to 8 g, which is eight times more gravity than on Earth. However, the physiological effect seems more important in planarians maintained at 8 g, since in this case we do observe alterations in the number of proliferating cells (Figure 5). It must be mentioned that while in the present study we observe an increase in the proliferative rates in 8 g animals, in a previous study we observed a decrease [27]. An explanation could be that the samples did not correspond to the same experimental approach, since in the present report we analyzed intact animals and in the previous one regenerating animals were studied.

Regarding the specific deregulated genes at s-µg, the down-regulation of cytoskeleton and matrix genes—such as *collagen-α-1* – is observed, a feature reported in several studies from other animal species [28]. Since the extracellular matrix is essential in maintaining the environment for stem cells, those defects could impact on the proper functioning and renewal of this population [29]. In agreement with this result, s-µg exposed animals also showed a down-regulation of *piwi* genes, which are essential genes to maintain the stemness, and in planarians are required for proper regeneration and tissue renewal [21]. Thus, although we observe a correct regenerative process, the genetic changes caused by gravity fluctuations could lead to severe consequences after long term exposure. It is surprising that, in planarians regenerating at 8 g, we do not find significant alteration of cytoskeleton or matrix proteins, since supporting higher mechanical forces requires strengthening the cell cytoskeleton to maintain the shape and function. 

Importantly, in planarians maintained at s-µg, the number of genes involved in ribosome biogenesis is particularly up-regulated, implying an increase in the number of ribosomes available to the cell, and therefore the cell translational capacity is increased. The deregulation of ribosomes levels and function is itself a mechanism involved in carcinogenesis [30]. Furthermore, the increase in the ribosomal content is a hallmark of cancer cells, since it allows the maintenance of a higher proliferation rate [22]. Thus, these results suggest that at s-µg cells could be sensitized to proliferative signals, and eventually predisposed to cancer. For this reason, we analyzed the proliferative levels in planarians maintained at s-µg conditions and in which the Hippo signaling was silenced. The rationale is that since Hippo inhibition promotes tumorogenesis through increasing stemness and proliferation, the exposure of those animals to altered gravity could potentiate the proliferative response. Furthermore, Hippo signaling is a conserved pathway that senses the physical properties of the environment to activate or repress the transcriptions of genes involved in proliferation, cell death or differentiation [31]. Since the sole inhibition of Hippo leads to over-proliferation, the alteration of gravity and thus of mechanical forces on cells could produce an additional disarrangement on the control of proliferation. Our results showed that in fact proliferation is increased in *hippo (RNAi)* animals submitted to s-µg conditions with respect to the *hippo (RNAi)* animals maintained at 1 g. This data clearly supports the idea that changes in gravity could be an additional factor predisposing the organism to illness. We also observe that *hippo (RNAi)* animals maintained at 8 g show an increase in the proliferation rates with respect to controls. The results show that the genetic alterations caused by gravity changes potentiate the proliferation in sensitive animals, such as in animals with silenced Hippo pathway. Several studies have shown inhibitory effects of microgravity on cancer cells viability and proliferation, which has lead to the proposal that microgravity could be a tool for exploring new anticancer therapies [32,33]. However, these studies have been developed in cancer cell lines, which already show multiple genetic defects. Our results indicate that the studies of *in vivo* and healthy systems, as planarians, are required to further understand the complex impact that altered gravity could have on living beings. For instances, it has also been reported that microgravity results in alteration of immune responses, which are essential to fight against cancer cells [33].

Overall, our results indicate that the exposure of plastic animals like planarians to gravity fluctuations produces important genetic modifications that contribute to cellular disarrangements, which are strengthened in a genetically sensitized background. Here, we show that genetic and cellular disarrangements are observed not only in cultured cells, as shown in numerous previous studies, but also in an *in vivo* complete organism context, which may resort to its own organismal compensation mechanisms. In the present study, the exposure of planarians to gravity fluctuations is not so long as the time period that astronauts spend in space. However, our data reinforce the necessity of future research investment in studying the genetic deregulations that humans could suffer during space missions, since these alterations could not only cause malfunctions on their own but could synergize or sensitize pre-existing alterations. Finally, it must be stressed that, since our research was carried out in simulated microgravity via the RPM, the effects reported here should be confirmed under real microgravity conditions.

## 4. Materials and Methods

### 4.1. Planarian Culture

Asexual planarians from a clonal strain of *Schmidtea mediterranea* BCN-10 were maintained at 20 °C in planarian artificial medium (PAM) water, as previously described [34]. Animals were fed with veal liver and starved for at least one week before beginning the experiments. Animals were transported from Barcelona/Spain to ESA-ESTEC Noordwijk/the Netherlands in PAM water using 50 mL falcons.

### 4.2. Simulations

Simulations were performed in ESTEC (ESA). Planarians were loaded into 15 mL falcons in PAM water at a density of 12 animals per falcon. Animals were kept in the dark and at a constant temperature of 21 °C during all the simulations. The Random Positioning Machine (RPM) was used simulate microgravity and the Large Diameter Centrifuge (LDC) was used to apply hypergravity (8 g). The Random Positioning Machine (RPM) to simulate microgravity was set to real random mode and random direction with a maximum speed of 10°/s. The samples were fixed in the center of the inner frame with a resulting largest radius of 9 cm to the outermost sample resulting in a maximum residual g due to rotation of less than 10^−4^ g [35]. RPM controls were placed in the same environment as the RPM. The LDC controls were placed at the center of the centrifuge in order to expose the samples to the same angular motion but without the additional gravitational load. Animals loaded in LDC were exposed to 8 g. Intact animals were loaded in the RPM and LDC at day 0. One day after loading, planarian heads and tails were amputated, and trunk fragments were re-loaded in the same devices (Figure 1). For candidate genes validation, planarian head fragments were loaded in the RPM using the previous settings. At 6 days after loading animals were fixed for in situ hybridization analysis ([36] Figure 4A). Animals at the RPM were imaged each 3–4 days without stopping the device. Animals at the LDC were not monitored to avoid stopping the centrifuge. Before RNA extraction or animal fixation each tube from the RPM experiment was examined in order to find possible air bubbles, which would have disturbed the applied forces.

### 4.3. RNA Interference Experiment

dsRNA was synthetized by *in vitro* transcription (Roche) and microinjection performed, as previously described [37], following the standard protocol of 3 × 32 nL injection of *hippo* dsRNA (1000 ng/μL) for three consecutive days. Control animals were injected with dsRNA for green fluorescent protein (GFP), a gene not present in planarians. For a detailed study about *hippo* silencing please see [24]. After the inhibition, animals were loaded in the RPM or in the LDC using the previous settings. Ten days after the animals’ loading, they were fixed and processed as described in the corresponding section.

### 4.4. Transcriptomic Analysis

For transcriptomic analysis, total RNA was extracted from our samples after 6 and 13 days of gravity fluctuations. Two replicates containing a pool of four planarians were analyzed for each condition. RNA was extracted with Trizol (Invitrogen, Carlsbad, CA, USA), following the manufacturer’s instructions. RNA was quantified with a Nanodrop ND-1000 spectrophotometer (Thermo Scientific, Waltham, MA, USA). In total, 16 libraries were prepared using the TruSeq RNA Library Prep Kit for single-end reads and sequenced in a HiSeq 2000 sequencer to obtain a minimum of 30 million reads per sample at the Genomics Unit facilities of the Centre for Genomic Regulation (CRG) and primary data processing was carried out by the Bioinformatics Unit. Sequence quality was assessed with FastQC [38] and sequences were also scanned for presence of rRNA contamination using riboPicker [39]. Since no reference genome is available for this species, the comprehensive transcriptome meta-assembly by Kao et al. [40] was used as reference for the read alignment. Filtered reads were aligned with Bowtie 2 [41] using the parameters to perform the posterior transcript assignation with eXpress [42] every multi-mapping was kept because of transcripts sharing the same sequences, penalties were increased for opening and extending gaps within reads or target regions, and modified function for calculating the minimum alignment score needed for an alignment to be considered valid. For the estimation of reads effectively mapping to a given transcript, eXpress quantifies the abundance of a set of target sequences by using an expectation–maximization algorithm for resolving ambiguous mappings. The algorithm estimates the number of fragments generated from each target in the sequencing experiment, adjusting for fragment and length biases. Following the instructions of the authors, we used the rounded “expected number of reads” as input for the statistical analysis with DESeq2 [43].

For the differential expression analysis, effective read counts were normalized by the total number of mapped reads and log-transformed with DESeq2 to minimize the bias, especially on lowly expressed genes, before the differential expression analyses. DESeq2 uses a model based on the negative binomial distribution for modeling biological variations and for detecting differentially expressed genes. Since all the controls corresponding to 12 days of regeneration clustered together in the PCA (Figure 2B) revealing thus a high similarity between them, they were considered as four replicates in the differential expression comparisons in order to increase the statistical power. The transcript fold change between conditions and controls and the corresponding adjusted p-value for multiple testing were estimated as listed in the Appendix A.

GO Enrichment analysis was performed using the Fisher’s exact test by comparing the GO terms present in the up- and down-regulated sets of genes obtained from the differential expression analysis with the full list of annotated transcripts in the transcriptome of reference using the Blast2GO software (v5.2.5, Biobam, Valencia, Spain) [44].

The RNA-Seq data generated for this publication have been deposited in NCBI’s Sequence Read Archive (SRA) [45] and are accessible through SRA BioProject accession number PRJNA516591 (http://www.ncbi.nlm.nih.gov/bioproject/516591).

### 4.5. In Situ Hybridization

RNA probes were synthesized *in vitro* using Sp6 or T7 polymerase (Roche, Basel, Switzerland) and DIG-modified (Perkin Elmer, Waltham, MA, USA) nucleotides. RNA probes were purified and precipitated with ethanol and 7.5 M ammonium acetate. Animals were fixed and processed as previously described [36]. Briefly, planarians water was replaced with 5% NAC solution, 5–10 min, room temperature (RT) and then animals were fixed with 4% formaldehide, 15–20 min at RT. Fixative was removed and animals were rinsed with PBSTx at RT. Then planarians were treated with preheated Reduction solution (50 mM DTT, 1% NP-40, 0.5% SDS, in 1× PBS), 5–10 min, at 37 °C. After removal of Reduction solution, animals were dehydrated with PBSTx—50% Methanol solution—100% Methanol, for 5–10 min each, at RT. Finally, methanol was replaced with 6% Bleach solution, under direct light, overnight, at RT. The next morning, animals were washed with PBSTx and bleached for two hours in formamide bleaching solution (1.2% H_2_O_2_, 5% formamide, and 0.5× SSC). Then, animals were washed with TNTx. Blocking was performed with 5% horse serum in TNTx. Antibodies were diluted in blocking solution for 16 h at 4 °C. The next morning post-antibody washes were done with TNTx. Then antibody was developed using 5-bromo-4-chloro-3-indolyl phosphate/nitro blue tetrazolium (BCIP/NBT) system. After staining, animals were mounted with 70% glycerol in PBS and observed.

### 4.6. Immunostaining and PH3 Quantification

Immunostaining was performed as previously described [46]. Briefly, animals were killed in cold 2% HCl in PBS for 4 min at RT. Animals were then fixed for 15 min at RT in 4% FA in PBSTx. Animal bleaching was performed overnight in 6% H_2_O_2_ diluted in PBSTx, at RT. The next morning, animals were washed twice with PBSTx before blocking for two hours at RT in PBSTB. Rabbit anti-phospho-histone-H3-Ser10 (anti-PH3) (1:500; Cell Signaling Technology) was diluted in PBSTB and incubated overnight at 4 °C. The following day, animals were washed extensively with PBSTx, then blocked in PBSTx with 0.25% BSA for one hour at RT. Then animals were incubated overnight at 4 °C in mouse anti-rabbit Alexa 568 secondary antibody. In the next day, animals were washed extensively with PBSTx and nuclei were stained with DAPI (1:5.000) and mounted with 70% glycerol in PBS and observed. To avoid technical variance and a reliable quantification of PH3+ cells, at least two independent experiments were performed.

Immunostaining samples were imaged using a MZ16F stereomicroscope (Leica) equipped with a ProgRes C3 camera (Jenoptik, Jena, Germany). Images were then processed using Fiji and Illustrator CC (Adobe, San José, CA, USA) software. Brightness/contrast and color balance adjustments were always applied to the entire image. Quantifications were performed by hand using the “multi-point selection” tool of Fiji. Animal area quantification was performed using the oval brush selection tool of Fiji and then normalized to mm^2^. Results were averaged per group and significant differences determined by 2-tailed *Student t*-test.

## Figures and Tables

**Figure 1 ijms-20-00720-f001:**
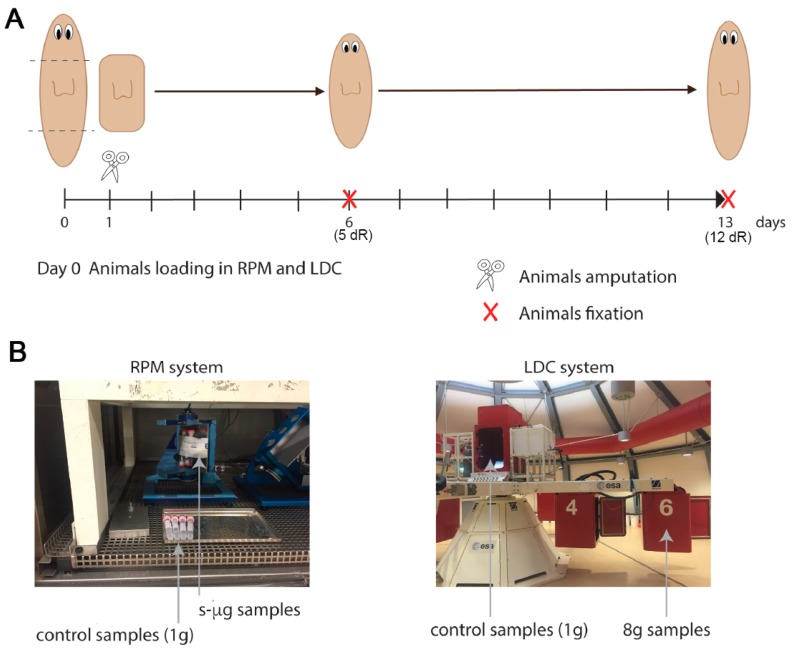
Illustration of the experimental design. (**A**) Animals were loaded to the Random Positioning Machine (RPM) or in the Large Diameter Centrifuge (LDC) at day 0. One day after the samples were removed and cut. Trunk fragments were loaded again until day 6 or day 13 (corresponding to day 5 or day 12 of regeneration, respectively). After exposure the animals were processed for RNA extraction. (**B**) Pictures of the RPM and LDC systems. The arrows indicate the placement of the samples during the experiment.

**Figure 2 ijms-20-00720-f002:**
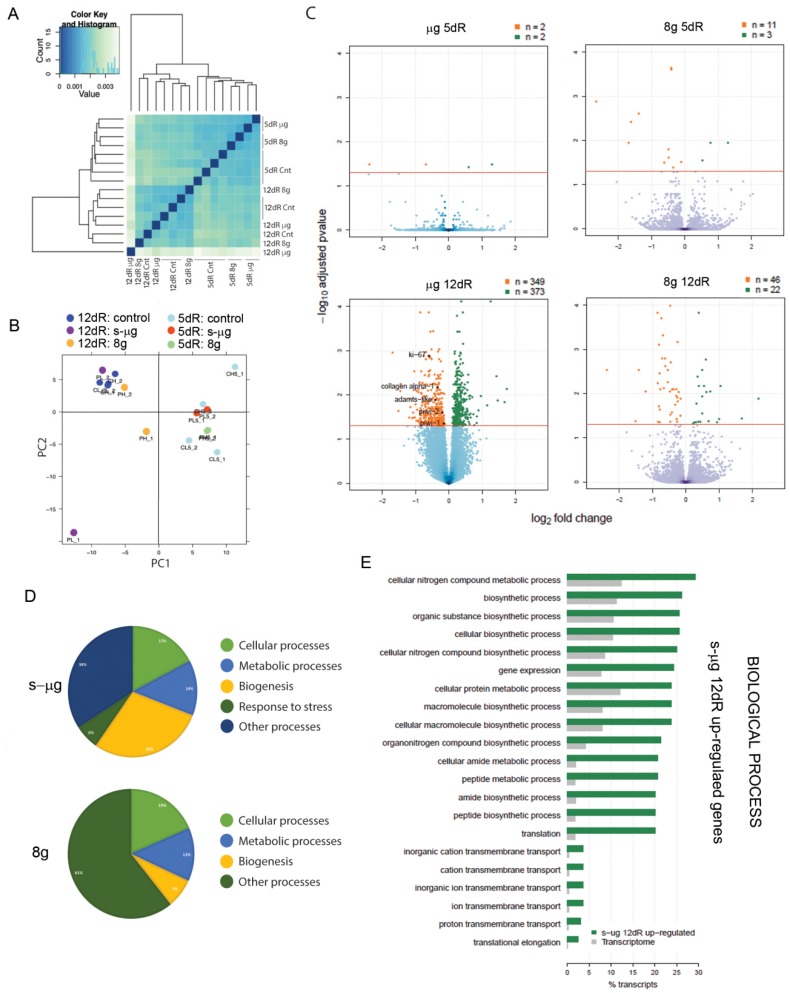
Transcriptomic analysis of samples exposed to s-µg or 8 g. (**A**) Histogram showing the distance between each pair of samples by Pearson correlation (**B**) Principal component analysis (PCA) of the samples, including the replicates for each condition: control (1 g), s-µg and 8 g at 12 dR and 5 dR. CL: control s-µg, 12 dR; CH: control 8 g, 12 dR; PL: s-µg, 12 dR; PH: 8 g, 12 dR; CL_5: control s-µg, 5 dR; CH_5: control 8 g, 5 dR; PL_5: s-µg, 5 dR; PH_5: 8 g, 5 dR. (**C**) Volcano plots showing the distribution of transcripts by expression fold change and significance. The red line indicates a significance level of 5% in the logarithmic scale of adjusted *p*-values. Significant up- and down-regulated transcripts (for and adjusted *p*-value < 0.05) are colored in green and orange, respectively, and their number is specified in the legend. Y-axis has been scaled to the same size in all the plots making some transcripts to be out of the scale and thus not shown here. (**D**) Gene ontology (GO) of differentially expressed genes in s-µg and 8 g. (**E**) GO Enrichment. Significantly (for a Fisher’s exact test FDR < 0.05) over- and under-represented GO functions corresponding to the s-µg samples after 12 dR in the up-regulated sets of differentially expressed genes as percentage of transcripts compared with the reference transcriptome. For a given function, if the percentage of sequences in the transcriptome is higher, that function is under-represented in the gene set.

**Figure 3 ijms-20-00720-f003:**
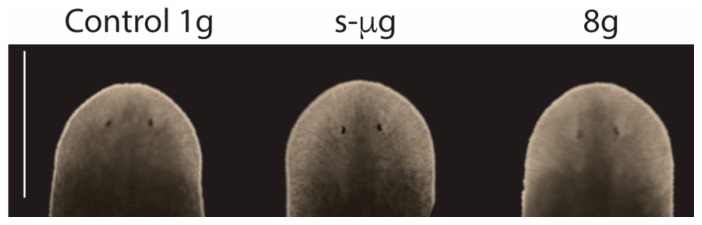
*In vivo* analysis of planarians regenerating at s-µg or 8 g. Head fragments of planarian trunk fragments in which the head had been amputated. They were regenerating s-µg or 8 g, as indicated. In all conditions they show an apparent properly regenerated head (note the eyes and the similar size of the blastema in the different animals). *n* ≥ 5. Scale bar: 150 µm.

**Figure 4 ijms-20-00720-f004:**
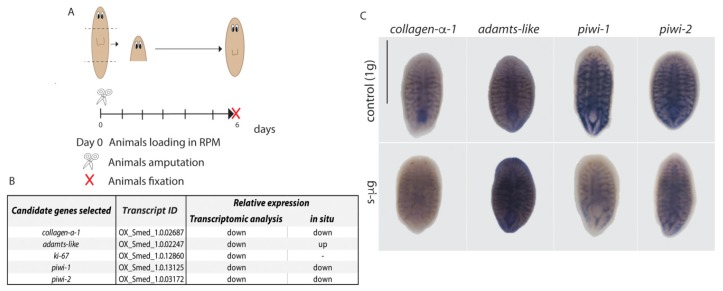
In situ hybridization validates the results of the transcriptomic analysis. (**A**) Diagram illustrating the experimental design followed to validate the transcriptomic results. Planarian head fragments were loaded into the RPM and animals were fixed and processed for in situ hybridization six days after. (**B**) Table showing the genes selected for validation. (**C**) Results of the in situ hybridization showing the down-regulation of *collagen-α-1, piwi-1*, and *piwi*-2 genes. *n* ≥ 5. Scale bar: 500 µm.

**Figure 5 ijms-20-00720-f005:**
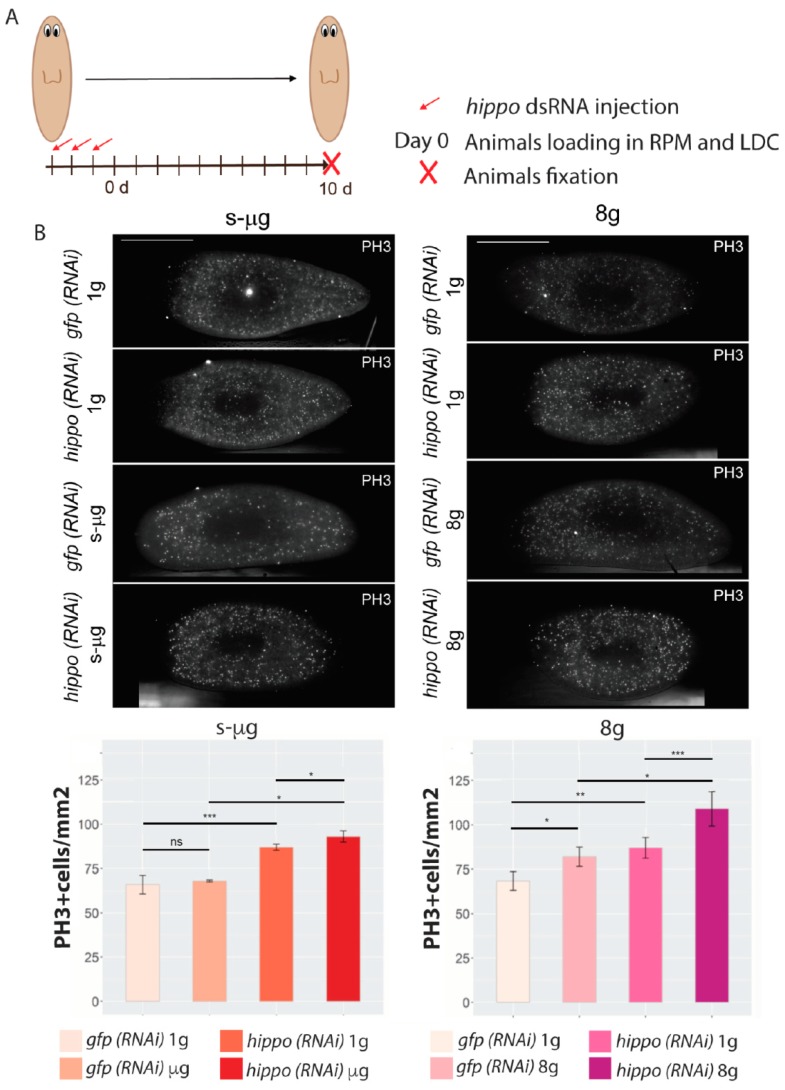
*hippo (RNAi)* planarians are more sensitive to the cellular effects of altered gravity. (**A**) Diagram illustrating the experimental design followed to analyze the effect of *hippo* inhibition in animals exposed to s-µg or 8 g. Intact animals were injected with *hippo* dsRNA during three consecutive days and then were loaded in the RPM or the LDC. After 10 days, animals were fixed and processed for immunofluorescence. (**B**) Analysis of PH3+ cells. Representative images of animals from each condition immunostained with PH3 antibody and graphs showing the quantification of PH3+ cells in each condition. Scale bar: 400 µm. Data is presented as means ± SD, *n* ≥ 10. * *p* < 0.05; ** *p* < 0.01; *** *p* < 0.001, *ns, not significant.*

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
