# Peer review of "Transcriptomic Analysis of Planarians under Simulated Microgravity or 8 g Demonstrates That Alteration of Gravity Induces Genomic and Cellular Alterations That Could Facilitate Tumoral Transformation"

_ijms, 2019, doi:10.3390/ijms20030720_

Round 1
Reviewer 1 Report
Overall this is a nice study reporting an interesting dataset, and should be published. I only have a couple of issues to be addressed:
1) Hippo RNAi: given that the "Hippo pathway is a well-known signaling pathway that controls the transcription of genes related with cell cycle, cell death and differentiation", what was the patterning phenotype of the Hippo RNAi animals - were they completely fine, or showed any changes in morphology, size control, etc. - this should be mentioned.
2) I did not see clear information on # of replicates for RNASeq experiments. For example, in Figure 2B, I see only 2 dark purple (13dR ug) dots which are very far apart, and only one green (5dR 8g) dot - am I misreading this plot, or does it indicate 2 and 1 replicate of these conditions? The authors should mention the # of replicates of each in the Methods section on RNASeq, and perhaps discuss how they dealt with interpretation of repeats that appeared very differently (the purple).
Author Response
Overall this is a nice study reporting an interesting dataset, and should be published. I only have a couple of issues to be addressed:
1) Hippo RNAi: given that the "Hippo pathway is a well-known signaling pathway that controls the transcription of genes related with cell cycle, cell death and differentiation", what was the patterning phenotype of the Hippo RNAi animals - were they completely fine, or showed any changes in morphology, size control, etc. - this should be mentioned.
We published the phenotype of hippo RNAi animals in de Sousa et al. 2018 (Plos Biology 2018). After 3 weeks of hippo inhibition animals show overgrowths and areas of unpigmented cells (there is cell dedifferentiation). The molecular and cellular analysis demonstrated that in those animals cycling cells are arrested in M phase, that there is a decrease of apoptotic levels and that differentiated cells cannot maintain their differentiated fate (de Sousa et al. 2018). All those traits could underlie overgrowth formation and tumoral transformation. The animals in the present study show an increase in M phase cells (Figure 5), as expected, but they do not show overgrowths or any apparent phenotype yet, because hippo was only silenced for 10 days.
2) I did not see clear information on # of replicates for RNASeq experiments. For example, in Figure 2B, I see only 2 dark purple (13dR ug) dots which are very far apart, and only one green (5dR 8g) dot - am I misreading this plot, or does it indicate 2 and 1 replicate of these conditions? The authors should mention the # of replicates of each in the Methods section on RNASeq, and perhaps discuss how they dealt with interpretation of repeats that appeared very differently (the purple).
It is true that this information was not clear in the previous version of the manuscript. We have now added this information in M&M sections. We also have added some explanation in the figure legend. We have analyzed two replicates for each condition. There are two green dots, but they overlap. There are four control dots for 5d and four for 12d. The reason is that we designed the experiment with two controls for each condition. They can be traced because of their name: CL refers to control low gravity and HL refers to control high gravity. However, for the samples corresponding to 12d we analyzed the four controls together, because the PCS showed they are very similar, and it increased the statistical power. This is now explained in detail in the text. Regarding the two purple dots, corresponding to 12dR animals at simulated microgravity, it is true that they are very far away in the PCA. However, we proceed with the two samples, since we did not have a third one. Of course, it would be nice to repeat this transcriptomic analysis, may be including other variables, to corroborate and improve the presented results.
Reviewer 2 Report
This manuscript reports the effects of both simulated microgravity and hypergravity on the transcriptome of planarians. It is demonstrated that microgravity induces cellular alterations at the transcript level at a higher extent than hypergravity condition and that Hippo silencing plays a role in cell proliferation as a response to altered gravity. Although this study presents interesting results with potential impact even for biomedical research, it is my opinion that the manuscript should not be published in the present form and needs major revisions before it can be reconsidered for publication, as follows:
Abstract
This section, as a standalone element of the manuscript, does not clearly state the message of the work and can be improved.
1- The rationale for using planarians is not clear. Authors first mention the lack of understanding about gravity alterations in human body and, immediately after, describe the experiment performed using planarians. Authors may just add that planarians were used as animal model of regeneration.
2- The last sentence of the abstract is not clear. It is evident from the results that “changes in gravity severely affect genetic transcription”, but what do authors mean when saying “these alterations potentiate molecular mechanisms that could promote the development of multiple diseases”?
Introduction
3- What do authors mean with “sensitized backgrounds”? From the experiments, one can understand that this may be referring to Hippo silencing, but in the context of the Introduction, this should be made clearer.
Results
4- Topic 2.1, as for simulated ug using the RPM, it should be mentioned that LDC was used to simulate hypergravity.
5- Presentation and description of results can be confusing. Sometimes authors refer to 5 and 12 dR, other times it appears 5 or 6dR and 13 dR. Although the experimental setup is clear, authors should chose only one designation for each condition.
6- In Figure 2, what do authors mean by “problem condition”? In fig2C, if red and blue dots represent up- and down-regulated genes, why is there a legend for 5dR and 13dR with the same color code? Are 5dR results presented in the upper part of the graphs and 13dR results in the lower part? If so, why is the 13dR represented with negative values for fold change? A volcano plot showing significance versus fold change could be easier to understand. Maybe transcript alterations upon exposure to either s-ug or 8g occur in the same genes, i.e., there might be some common changes to both conditions. Is this happening? If so, perhaps a venn diagram could be useful to illustrate this.
7- In topic 2.1, authors state “We observed two main groups that correspond to the two regeneration stages of planarians – 5 or 12 days of regeneration. In addition, through principal component analysis (PCA) of our samples we observed that they are mainly correlated with regeneration stage – either 5 (5dR) or 12 days of regeneration (12dR) – and with condition – s-g or 8g (Figure 2B).” However, it is not clear from the graphs that differences between condition represent different groups. A table (or tables) summarizing the data could also be useful- maybe as supporting information. It would be interesting to see whether different transcripts were differentially found (up- or down-regulation) between conditions and between different dR. This could be organized by number of transcripts in rows (total up-regulated; up-regulated in 5dR, up-regulated in 13dR; total down-regulated, down-regulated in 5dR, down-regulated in 13dR) versus transcriptome comparisons in columns (s-ug vs ctr; 8g vs ctr; s-ug vs 8g), for instance, depending on what makes sense to show here for the complete data.
8- Authors compare the obtained results from this collective transcriptomic analysis with previous data, stating “correct regeneration of the structures and no change in proliferation levels on these conditions”. It is not clear how this can be extrapolated from the analysis presented in fig2. What are the relations between altered transcripts and regeneration or proliferation? This can’t be concluded from data on fig.2 only. This should be included in the discussion section, rather than on results.
9- Can authors identify a list of genes (e.g., top 10) with highest changes in each condition?
10- In topic 2.3, lines 184-185, please mention that this is the result for hippo (RNAi) planarians when describing PH3 results at 8g. Fig 5: are there any statistical differences between “normal” animals and hippo silenced ones? Authors describe “hippo (RNAi) planarians are more sensitive to the cellular effects of altered gravity” but no comparison is shown between hippo (RNAi) s-ug (or 8g) and their gfp (RNAi) counterparts. To conclude that hippo planarians are more sensitive to altered gravity, results should be compared to those of gfp animals under the same condition, besides the differences between hippo 1g and hippo s-ug (or 8g).
Materials and Methods
11- In topic 4.2, should “RMP” be “RPM” instead? Were the environmental conditions (dark, temperature of 21oC) maintained during all the simulations? How was this monitored/controlled over the 13 days of the experiments? Add a note saying that LDC was used for simulating hypergravity conditions – this is just for readers from outside the field to better understand and to clarify that this is also simulated (as for RPM s-ug). It is not clear why reference 28 (van Loon, Adv Sp Res) is cited regarding in situ hybridization analysis – is it a mistake, maybe reference 29 should be cited? The same applies to reference 29- should be it be ref30 instead? Please mention “candidate genes”, rather than “candidate” only.
12- Description of experiments on Hippo silencing could be separated into a different M&M topic and more details could be added – also include here your own citation (ref22).
13- In topic 4.3, transcriptomic analysis was performed using assay triplicates or three experimental replicates? How many experiments were performed? Details on RNA extraction protocol and sample processing should be added.
14- Please detail the immunostaining protocol (fixation steps, incubation conditions, etc). How was PH3 quantification performed? How many cells were analyzed?
15- Statistical analysis description is missing. Details on the analysis of transcriptomics data should also be added to the M&M section.
Discussion
Overall, this section should be completed by discussing obtained results in light of what is reported in the literature.
16- In the first paragraph, authors refer that gravity alterations can result in cellular malfunctions (over-proliferation) and progress of some diseases like cancer. According to results presented in the study, this is only valid for hippo silenced animals. One can hardly conclude this, since no differences are shown between 1g and simulated gravity conditions for gfp (RNAi) animals. Also, do authors have control/unmodified animals for these experiments? If there is data on this (even from the literature), it could help understanding whether the silencing protocol itself affected planarians to some extent.
17- How are proliferation results comparable to previous study by the team (Adell et al. 2014, http://dx.doi.org/10.1155/2014/679672)? Although no differences have been found for s-ug (gfp/normal animals 1g vs ug), authors have previously reported a decrease in the number of PH3+ cells under hypergravity (both 4g and 8g). Is there any explanation for these opposite results? Can it be a consequence of the modification protocol itself? Authors should better discuss this.
18- Still referring to the same paragraph, authors should be more careful when extrapolating for cancer progression. Although this can be easily to understand for a biologist, readers from a different field may need some background on this. Can authors explain more in detail why there might be an association between the two events? This result is an interesting aspect with potential biomedical implications, but the rationale behind should be made clearer. Besides, when discussing implications for cancer, what is the rationale for using planarians for such experiments? There are some studies using cancer cells, including some that report antiproliferative and antimetastatic effects of real microgravity/spaceflight (https://doi.org/10.1096/fj.13-243287). For instance, Ma et al 2014 reports the following (among other results): “Our experiments indicate that microgravity, preferably real as in spaceflight, but also in some respects simulated on the RPM, induce changes in the expression and secretion of genes and proteins involved in cancer cell proliferation, metastasis, and survival, shifting the cells toward a less aggressive phenotype.”
How is this comparable? Authors should better address these differences in the discussion section.
19- Authors could address the role of Hippo signaling in cancer/regeneration/differentiation in more detail. Since they have previously hypothesized that “the main role of Hippo in planarians is not to promote proliferation but to control the cell cycle and maintain a stable differentiated cell fate” (ref22), what are the implications of this under altered gravity?
Minor comments:
- Keywords should be listed alphabetically
- LDC stands for “Large” instead of “Long Diameter Centrifuge”: this mistake appears at least in topic 2.1 and abbreviations list
- Fig3: clarify that these results are related to planarian trunks regenerating heads (instead of head fragments regenerating).
Author Response
Thanks for the comments. We answer them point by point as follows.
# Reviewer 2
This manuscript reports the effects of both simulated microgravity and hypergravity on the transcriptome of planarians. It is demonstrated that microgravity induces cellular alterations at the transcript level at a higher extent than hypergravity condition and that Hippo silencing plays a role in cell proliferation as a response to altered gravity. Although this study presents interesting results with potential impact even for biomedical research, it is my opinion that the manuscript should not be published in the present form and needs major revisions before it can be reconsidered for publication, as follows:
Abstract
This section, as a standalone element of the manuscript, does not clearly state the message of the work and can be improved.
1- The rationale for using planarians is not clear. Authors first mention the lack of understanding about gravity alterations in human body and, immediately after, describe the experiment performed using planarians. Authors may just add that planarians were used as animal model of regeneration.
2- The last sentence of the abstract is not clear. It is evident from the results that “changes in gravity severely affect genetic transcription”, but what do authors mean when saying “these alterations potentiate molecular mechanisms that could promote the development of multiple diseases”?
Thanks. We have rewritten the abstract accordingly.
Introduction
3- What do authors mean with “sensitized backgrounds”? From the experiments, one can understand that this may be referring to Hippo silencing, but in the context of the Introduction, this should be made clearer.
Yes, we have added an explanation.
Results
4- Topic 2.1, as for simulated ug using the RPM, it should be mentioned that LDC was used to simulate hypergravity.
We do not understand this concern. We already explain in the text that LDC was used to apply 8g. In fact, hypergravity is not simulated but it is real, since the force sensed by planarians in the LDC at the conditions used is exactly 8g.
5- Presentation and description of results can be confusing. Sometimes authors refer to 5 and 12 dR, other times it appears 5 or 6dR and 13 dR. Although the experimental setup is clear, authors should chose only one designation for each condition.
We have corrected the labels in Figure 2B. It was a mistake that may be lead to the confusion. In general we refer to the days of regeneration, but we have not removed the explanation of what corresponds in each case, in the first paragraph of the results, since we think that it clarifies the strategy.
6- In Figure 2, what do authors mean by “problem condition”? In fig2C, if red and blue dots represent up- and down-regulated genes, why is there a legend for 5dR and 13dR with the same color code? Are 5dR results presented in the upper part of the graphs and 13dR results in the lower part? If so, why is the 13dR represented with negative values for fold change? A volcano plot showing significance versus fold change could be easier to understand. Maybe transcript alterations upon exposure to either s-ug or 8g occur in the same genes, i.e., there might be some common changes to both conditions. Is this happening? If so, perhaps a venn diagram could be useful to illustrate this.
Red and blue dots did not represent up- and down-regulated genes but regeneration stage. However, and according to the reviewer comment, we have now substituted these graphics by a volcano plot showing significance versus fold change (new Figure 2C). We thank the reviewer comment, since the volcano plot helps to the understanding of the results.
On the other hand, we have checked whether might be common changes in s-ug or 8g conditions. There are very few coincidences, so we have not constructed a Venn diagram, but this was in fact an interesting comment.
In the next table we detail the number of genes that are found deregulated in two different conditions:
7- In topic 2.1, authors state “We observed two main groups that correspond to the two regeneration stages of planarians – 5 or 12 days of regeneration. In addition, through principal component analysis (PCA) of our samples we observed that they are mainly correlated with regeneration stage – either 5 (5dR) or 12 days of regeneration (12dR) – and with condition – s-g or 8g (Figure 2B).” However, it is not clear from the graphs that differences between condition represent different groups. A table (or tables) summarizing the data could also be useful- maybe as supporting information. It would be interesting to see whether different transcripts were differentially found (up- or down-regulation) between conditions and between different dR. This could be organized by number of transcripts in rows (total up-regulated; up-regulated in 5dR, up-regulated in 13dR; total down-regulated, down-regulated in 5dR, down-regulated in 13dR) versus transcriptome comparisons in columns (s-ug vs ctr; 8g vs ctr; s-ug vs 8g), for instance, depending on what makes sense to show here for the complete data.
We have now removed the first table, since this information is also included in the second table (now Figure 2B). In this table each sample is represented by a dot, and the color indicated the treatment of each one. However, it is true that this representation does not reflect differences between conditions, and, according to the reviewer comment, we have now added a table with the list of transcripts found up- or down- regulated in each condition (Supp table2). The transcripts are arranged by p-value. The annotation is showed (when it is possible).
8- Authors compare the obtained results from this collective transcriptomic analysis with previous data, stating “correct regeneration of the structures and no change in proliferation levels on these conditions”. It is not clear how this can be extrapolated from the analysis presented in fig2. What are the relations between altered transcripts and regeneration or proliferation? This can’t be concluded from data on fig.2 only. This should be included in the discussion section, rather than on results.
Yes, thanks, we have included a paragraph comparing the new data with the reported one in the discussion.
9- Can authors identify a list of genes (e.g., top 10) with highest changes in each condition?
Yes, now this information is included in Supp table 2
10- In topic 2.3, lines 184-185, please mention that this is the result for hippo (RNAi) planarians when describing PH3 results at 8g. Fig 5: are there any statistical differences between “normal” animals and hippo silenced ones? Authors describe “hippo (RNAi) planarians are more sensitive to the cellular effects of altered gravity” but no comparison is shown between hippo (RNAi) s-ug (or 8g) and their gfp (RNAi) counterparts. To conclude that hippo planarians are more sensitive to altered gravity, results should be compared to those of gfp animals under the same condition, besides the differences between hippo 1g and hippo s-ug (or 8g).
We have now included the statistical significance of all relevant comparisons. There is a significant increase in the hippo RNAi samples with respect to the controls at 1g, in both experiments, as expected according to de Sousa et al. 2018.
Materials and Methods
11- In topic 4.2, should “RMP” be “RPM” instead? Were the environmental conditions (dark, temperature of 21oC) maintained during all the simulations? How was this monitored/controlled over the 13 days of the experiments? Add a note saying that LDC was used for simulating hypergravity conditions – this is just for readers from outside the field to better understand and to clarify that this is also simulated (as for RPM s-ug).
All this information is now included in M&M section. Thanks.
It is not clear why reference 28 (van Loon, Adv Sp Res) is cited regarding in situ hybridization analysis – is it a mistake, maybe reference 29 should be cited? The same applies to reference 29- should be it be ref30 instead?
Please mention “candidate genes”, rather than “candidate” only.
We have fixed it.
12- Description of experiments on Hippo silencing could be separated into a different M&M topic and more details could be added – also include here your own citation (ref22).
We have included the detailed protocol in M&M.
13- In topic 4.3, transcriptomic analysis was performed using assay triplicates or three experimental replicates? How many experiments were performed? Details on RNA extraction protocol and sample processing should be added.
Thanks, a detailed explanation of the transcriptomic procedure has now been included in M&M.
14- Please detail the immunostaining protocol (fixation steps, incubation conditions, etc). How was PH3 quantification performed? How many cells were analyzed?
A detailed protocol of the pH3 immunostaining and its quantification has now been included in M&M.
15- Statistical analysis description is missing. Details on the analysis of transcriptomics data should also be added to the M&M section.
Statistical analysis description for the H3P quantification is added in the immunohistochemistry section of M&M. Details of the analysis of the transcriptomics data is added in the corresponding section. Previous Supplementary Figure 1 and Supplementary Table 2 have been removed, since we think that with the explanations added these data were not necessary. Thanks for the comment.
Discussion
Overall, this section should be completed by discussing obtained results in light of what is reported in the literature.
16- In the first paragraph, authors refer that gravity alterations can result in cellular malfunctions (over-proliferation) and progress of some diseases like cancer. According to results presented in the study, this is only valid for hippo silenced animals. One can hardly conclude this, since no differences are shown between 1g and simulated gravity conditions for gfp (RNAi) animals. Also, do authors have control/unmodified animals for these experiments? If there is data on this (even from the literature), it could help understanding whether the silencing protocol itself affected planarians to some extent.
We do not refer that directly gravity alterations lead to cellular malfunctions. We now indicate that our data shows that during long exposures or in a sensitized background, they could lead to transcriptional changes, cellular malfunctions (over-proliferation) and progress of some diseases such cancer.
17- How are proliferation results comparable to previous study by the team (Adell et al. 2014, http://dx.doi.org/10.1155/2014/679672)? Although no differences have been found for s-ug (gfp/normal animals 1g vs ug), authors have previously reported a decrease in the number of PH3+ cells under hypergravity (both 4g and 8g). Is there any explanation for these opposite results? Can it be a consequence of the modification protocol itself? Authors should better discuss this.
Yes, thanks. It is an important point that was missing in the previous version of the manuscript. We have not added an explanation in the discussion section.
18- Still referring to the same paragraph, authors should be more careful when extrapolating for cancer progression. Although this can be easily to understand for a biologist, readers from a different field may need some background on this. Can authors explain more in detail why there might be an association between the two events? This result is an interesting aspect with potential biomedical implications, but the rationale behind should be made clearer. Besides, when discussing implications for cancer, what is the rationale for using planarians for such experiments? There are some studies using cancer cells, including some that report antiproliferative and antimetastatic effects of real microgravity/spaceflight (https://doi.org/10.1096/fj.13-243287). For instance, Ma et al 2014 reports the following (among other results): “Our experiments indicate that microgravity, preferably real as in spaceflight, but also in some respects simulated on the RPM, induce changes in the expression and secretion of genes and proteins involved in cancer cell proliferation, metastasis, and survival, shifting the cells toward a less aggressive phenotype.”
How is this comparable? Authors should better address these differences in the discussion section.
Yes, it is in fact very interesting. Thanks. We have included a paragraph in the discussion.
19- Authors could address the role of Hippo signaling in cancer/regeneration/differentiation in more detail. Since they have previously hypothesized that “the main role of Hippo in planarians is not to promote proliferation but to control the cell cycle and maintain a stable differentiated cell fate” (ref22), what are the implications of this under altered gravity?
We have added a paragraph discussing the role of hippo in mechanotransduction.
Minor comments:
- Keywords should be listed alphabetically
- LDC stands for “Large” instead of “Long Diameter Centrifuge”: this mistake appears at least in topic 2.1 and abbreviations list
- Fig3: clarify that these results are related to planarian trunks regenerating heads (instead of head fragments regenerating).
Thanks, we have corrected it.

Reviewer 3 Report
In this paper, the impact of micro and hyper gravity on planaria is explored. The authors claim that gravity fluctuations induce transcriptomic alterations that could lead to the development of cancer.
In my opinion there two major relevance problems. (1) gravity impacting human health during space trips is not well-formulated in the experimental design of the study. Astronauts spend prolonged periods of time in space under microgravity. The length of the present experiments certainly do not exceed these times and as such they can only monitor relatively short-term effects on animal biology. (2) it is not well understood why the authors chose 8g for hyper-gravity. The most gravity experienced by humans is ~3.2xg during launch of a spaceship. The gravity of planets other than earth is between ~2.6xg and 0.2xg.
In the technical aspects of the paper:
- RNA-seq has been properly processed.
- GO analysis is very basic. Even though what is reported is correct, it does not provide any meaningful biological significance. What I think is going to make the paper better is for the authors to provide an enrichment analysis of each comparison presented.
- Authors used the head fragments to validate their RNA-seq data but, as they have also noted (line 128-9), it is not the tissue used for the RNA-seq experiments, so I do not believe that it is an appropriate experiment to validate the data. I think authors should consider using qPCR on the same tissue that was used for the RNA-seq experiment in order to properly validate their data.
- The link between the RNA-seq data and exploring Hippo signaling is not well founded. Metabolic reprogramming is a cancer hallmark but that is not necessarily reflected on GO terms. The GO term metabolic process is very generic and represent almost 5,500 genes in the human genome. There are specific metabolic reprogramming events that make a cell cancerous.
- It would make the paper better if there are representative pictures of the pH3 staining for Figure 5.
- The quantification in Figure 5 lacks what statistics were used to obtain the given p-values. In addition, the authors mentioned that there are no differences in proliferation rate when planaria experience micro or hyper gravity (ref 14), but gfp (RNAi) 1g and gfp(RNAi) 8g data appear significant, even more since hippo(RNAi)1g and hippo(RNAi)ug are labeled as significant. The data do not support the claim that gravity fluctuations intensify proliferation in a genetically unstable background (rephrased from line 186-7). My opinion is that the authors should also explore the possibility that 8g damages cells and then proliferation is increased to replace them. Also see my comment above for the use of 8g.
Author Response
# Reviewer 3
In this paper, the impact of micro and hyper gravity on planaria is explored. The authors claim that gravity fluctuations induce transcriptomic alterations that could lead to the development of cancer.
In my opinion there two major relevance problems. (1) gravity impacting human health during space trips is not well-formulated in the experimental design of the study. Astronauts spend prolonged periods of time in space under microgravity. The length of the present experiments certainly do not exceed these times and as such they can only monitor relatively short-term effects on animal biology. (2) it is not well understood why the authors chose 8g for hyper-gravity. The most gravity experienced by humans is ~3.2xg during launch of a spaceship. The gravity of planets other than earth is between ~2.6xg and 0.2xg.
Thanks for the comments. Regarding the first one, it is true that astronauts spend longer periods of time, but we just aimed to approach the situation. We performed a 13 d experiment because if a single animal dies during the experiment, then all animals in the same tube also would die. So, we preferred to limit the experimental period but being sure that all animals are healthy. In the discussion, we have added a clarification that of course the duration of our experiment is shorter than the real period that astronauts spend in space. On the other hand, it also must be considered that each animal could sense the environmental changes in a different manner. So, the time period required for a planarian to translate changes in gravity in genetic changes may be is not the same for a human. So, ours is just an experimental approach that allows concluding that in a whole animal, gravity fluctuations impact on gene expression, although phenotipically we observe no changes.
Regarding the second comment, in previous experiments we tested the effect of 3g, 4g and 8g in planarians (Adell et al. 2014). The results were that planarians could regenerate properly in any of these conditions. This was one of the reasons to use 8g, as a gravity challenge to explore the impact of weight on a system. This is not much different than to use a range of concentrations when exploring e.g. impact of growth factors or other challenges. Furthermore, in the previous experiment we also observed that planarians at 4g, but not at 8g, fission at high rates. Fission of the tail is the natural mode of reproduction in asexual planarians. It is not well understood, but planarians fission the tail in response to some environmental changes. The fact is that, we do not know the reason but planarians at 4g suffered high rates of fission, but not planarians at 8g. This was an essential information at the time of choosing the experimental condition, since it is not possible to perform a comparative transcriptomic approach comparing control animals (not fission) with planarians at 4g that suffered fission, since they are regenerating the missing structures and the transcriptomic data would not be comparable.
In the technical aspects of the paper:
- RNA-seq has been properly processed.
- GO analysis is very basic. Even though what is reported is correct, it does not provide any meaningful biological significance. What I think is going to make the paper better is for the authors to provide an enrichment analysis of each comparison presented.
Yes, thanks for the comment. We now have performed a GO enrichment analysis and we provide the results in the new Figure 2E and in the new Supp Fig 1.
- Authors used the head fragments to validate their RNA-seq data but, as they have also noted (line 128-9), it I think authors should consider using qPCR on the same tissue that was used for the RNA-seq experiment in order to properly validate their data.
It is a possibility, but in fact we find more convincing to validate it in a different fragment type, since the result is more robust. If we find downregulation of a specific gene in the transcriptomic analysis of trunk fragments and also in an in situ experiment of a head fragment, then it means that the result in really robust. If we find the same downregulation in the same type of fragment and by qPCR, in fact it would be just like an additional replicate of the transcriptome.
- The link between the RNA-seq data and exploring Hippo signaling is not well founded. Metabolic reprogramming is a cancer hallmark but that is not necessarily reflected on GO terms. The GO term metabolic process is very generic and represent almost 5,500 genes in the human genome. There are specific metabolic reprogramming events that make a cell cancerous.
Yes, thanks for the comment. We have now analyzed more in detail the genes found de-regulated specially at s-mg, and we found that there are several ribosomal proteins and other genes related with protein translation. We have now discussed this finding in the text. In supp Table 3 and in Supp Fig 1 we now show the results of the GO enrichment analysis, which further corroborates these results.
- It would make the paper better if there are representative pictures of the pH3 staining for Figure 5.
Yes, we have added them in Figure 5. Thanks.
- The quantification in Figure 5 lacks what statistics were used to obtain the given p-values.
The statistical procedure has been detailed in M&M section.
In addition, the authors mentioned that there are no differences in proliferation rate when planaria experience micro or hyper gravity (ref 14), but gfp (RNAi) 1g and gfp(RNAi) 8g data appear significant, even more since hippo(RNAi)1g and hippo(RNAi)ug are labeled as significant.
Thanks for the comment. We have added in the graphs the statistical significance of all comparisons that are reliable in this experiment. Regarding the mg data, the results are clear: 1) we find no difference between control and mg samples, which is the same result reported in Adell et al 2014; 2) we do find an increase in proliferation in hippo RNAi animals with respect to the controls, both at 1g, which is also consistent with the results found in de Sousa et al. 2018; 3) the most important, we do find an increase in the proliferation levels in hippo RNAi animals at mg both with respect to controls at mg and to hippo RNAi at 1g. This data supports our hypothesis that the simultaneous inhibition of hippo and exposure to mg has a synergistic effect. This result is also found in animals exposed to 8g: 1) hippo RNAi animals exposed at 8g show an increase in proliferation with respect to hippo RNAi animals at 1g and to control animals at 8g; 2) in this experiment we also find an increase in proliferation when inhibiting hippo in 1g samples. However, it is true that in this experiment we find an increase in proliferation in animals exposed to 8g with respect to the control animals at 1g, which is the opposite result found in our previous report (Adell et al 2014).The reason could be that in Adell et al 2014 the samples analyzed were trunks at 12 days of regeneration, and in the present study the samples are intact animals. In any case, in both studies the significance is not very high (>0,01). We have now corrected the interpretation of the 8g data according to the results published in Adell et al. 2014, both in the results and in the discussion.
The data do not support the claim that gravity fluctuations intensify proliferation in a genetically unstable background (rephrased from line 186-7). My opinion is that the authors should also explore the possibility that 8g damages cells and then proliferation is increased to replace them. Also see my comment above for the use of 8g.
It is a possibility that in 8g samples cells are damaged and proliferation is increased, but in any case, we find a very significant increase in the proliferative rates when 8g animals also have hippo silenced. This means that, even if the reason is damage in cells, the fact is that the simultaneous application of both parameters (hippo RNAi and hypergravity) produces an increase in proliferation that is higher than in any of both conditions alone. Furthermore, this result is also found in mg conditions, which further supports our claim that gravity fluctuations intensify proliferation in a genetically unstable background. It could be that at 8g cells are damaged, but we consider that this possibility does not change the conclusion. It would be interesting to check cell death in those animals. However, we must stress that 8g animals really did not looked unhealthy, and did not present any sign of necrosis, which is very easy to distinguish in planarians. In any case, it is a possibility that could be further explored.
Round 2
Reviewer 2 Report
The authors addressed all the comments. The manuscript has been improved.
Reviewer 3 Report
In their revised version, authors have considerably improved the manuscript. I still have the following concerns that in my opinion are important:
- The validation of the RNA-seq experiment using an additional method that measures RNA expression, like qPCR, is fundamental for rigorous scientific experimentation. Using different fragments of the planaria, does not replicate the experiment, thus, cannot validate it. Authors should perform an identical experiment for validation of the RNA-seq experiment in order to fully support their claim of "validated" RNA-seq.
- Several claims in the manuscript, including the title, need to become softer. For instance in the title: "Genomic alterations" should become "transcriptomic alterations", there are no "cellular" alterations other than proliferation defects, and there is no formation of tumors. So currently, the title does not reflect the paper.
- The experimental data supporting a link between cancer and gravity are weak. The text should be majorly revised.
- Authors should consider discussing this paper: Planarian regeneration in space: Persistent anatomical, behavioral, and bacteriological changes induced by space travel Morokuma J., et al. 2017.